# Global Art Market in the Aftermath of COVID-19: A Case Study on the United Arab Emirates

**Eve Grinstead**

École Normale Supérieure (Ulm), Université Paris Sciences Lettres (PSL), ED 540—École Doctorale Lettres, Arts, Sciences Humaines et Sociales, L'Institut d'histoire Moderne et Contemporaine (IHMC), UMR 8066, 75005 Paris, France; evegrinstead@gmail.com

**Abstract:** How has COVID-19 affected the global art market? This virus interrupted 2020 in unforeseen ways globally, including the cancellation of the most important art events of the year. Through a close chronological study of the Emirati art scene's response, both in commercial and noncommercial venues, this essay explains how, and why, the UAE's art scene was able to react quickly and perhaps more effectively than that of other nations, and what that means for its future. Based on fieldwork and press articles, this article posits that the Emirati art scene evolved from being virtually non-existent to a thriving contemporary art hub in a matter of decades because it has always had to adapt to challenges such as nonexistent art infrastructure or the 2008 financial crisis. By studying the UAE, we find examples of exhibitions that quickly moved from being in situ to online, a rare instance of galleries and art auction house collaborating, government and institutional structures stepping up to support artists and galleries, and the renaissance of Art Dubai taking place in person in 2021 after being abruptly cancelled in 2020. This knowledge provides insight into how the global art market is changing to face the consequences of COVID-19.

**Keywords:** United Arab Emirates (UAE); Art Dubai; Alserkal Avenue; Sotheby's Dubai; virtual exhibitions; Abu Dhabi Art; post-COVID-19 art market



## 1. Introduction

COVID-19 interrupted nearly every industry internationally in 2020, and in some places still does. The United Arab Emirates (UAE) was, however, less affected than other locations, which is evident in its contemporary art scene. Due to the quickly growing nature and innate adaptability of this market—whose epicentre is Dubai—contemporary art in this small country was able to not merely survive but prosper during the pandemic. It should be noted that the sources for the arguments put forth in this essay are largely based on fieldwork and press articles, not theoretical academic research. This is a conscious choice for two reasons: first, to not reveal longer-term research currently underway, and second, since this article addresses a current phenomenon that is still ongoing, the decision was made to give greater importance to historical context, in addition to providing a highly detailed account of the Emirati art scene's reaction to COVID-19—and how they are related—since this collective behaviour is precisely what we posit has helped it fare well through the pandemic.

As a brief history, the UAE (see Figures 1 and 2) became a federation of seven states—Abu Dhabi, Dubai, Sharjah, Umm al-Qaiwain, Fujairah, Ajman, and Ras al-Khaima—in 1971 (Heard-Bey 1982). Its art community emerged only a few years later, in the late 1970s and early 1980s, through a variety of structures, both commercial and not-for-profit sectors. In 1979, Sheikh Dr. Sultan bin Muhammad Al Qasimi, the ruler of Sharjah, declared a 'Revolution of Culture' for his emirate, which placed an emphasis on the fine arts and higher education, established a book fair, and a theatre (Kazerouni 2017). That same year, British expatriate Allison Collins founded Dubai's first art gallery—Majlis—in her home in Bastakiya (Collins 2020). Throughout the gallery's first decade, Collins hosted many shows

without an official license, but it became officially registered as an LLC in 1989 (Moghadam 2012). The Emirates Fine Arts Society (EFAS) opened in 1980 in Sharjah to host classes and annual exhibitions for its members and invited artists.

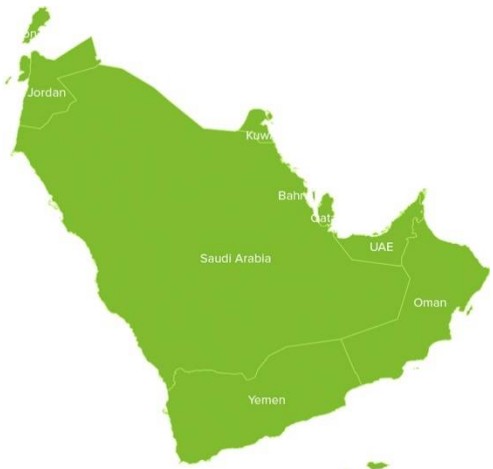

**Figure 1.** Map of the Arabian Peninsula illustrating the UAE in relation to neighbouring countries. Used with permission from Pngegg.com accessed on 1 June 2021.

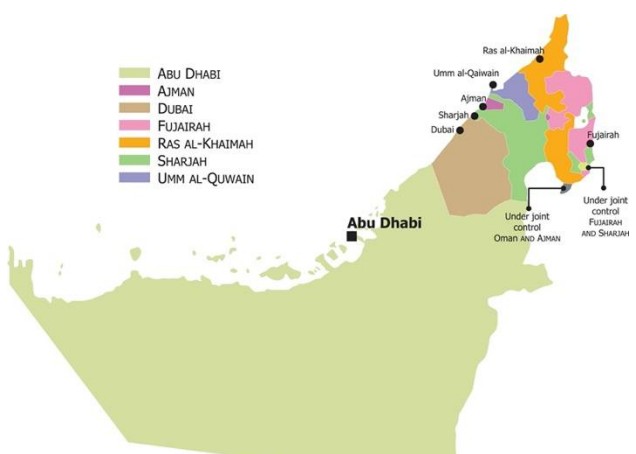

**Figure 2.** Map of the seven states of the United Arab Emirates. Used with permission from Pngegg.com accessed on 1 June 2021.

A year later, Abu Dhabi's Cultural Foundation opened as an arts and culture centre and hosted the first national library. In 1987, the 'father' of the Emirati art scene, Hassan Sharif, founded the Dubai Art Atelier (Allison 2017), and since the 1990s, residencies such as those at the Khor Fakkan Public Library (founded in 1990) and EFAS (1992) were launched, establishing a long tradition of artistic support: Today, those at Tashkeel (founded in 2008), Alserkal Avenue (founded in 2015) and the Sheikha Salama Foundation (Salama Emerging Artist Fellowship, SEAF, founded in 2013) in Abu Dhabi continue to nourish the arts. In 1993, Sharjah remained at the forefront of the noncommercial scene by establishing the Sharjah Biennial, and two years later, the Sharjah Art Museum. Since 1997, the UAE French Cultural Centre has hosted exhibitions that showcase the local art scene, such as *UAE Artists* and *Emirates Identities* in 2000 (Allison 2017).

In the mid-1990s, the current commercial art landscape began to take form. In Abu Dhabi—known less for its galleries and more for large, international, and institutional partnerships—the Salwa Zeidan Gallery opened in 1994. In Dubai, the following year, the Green Art Gallery, established in 1987 in Homs, opened another branch in a villa in the

residential neighbourhood of Jumeriah; in 1996, the 1 × 1 Gallery in Satwa (Dubai) was founded, and between 1997 and 1998 the Iranian architect Dariush Zandi opened the Total Arts Gallery in Al Quoz—one of the first galleries in this industrial district. Joining Zandi in the area, Sharon Harvey launched the Showcase Gallery at this time; this district is now the centre of Dubai's most reputable galleries. In 1998, Alanood Al-Warshow, the first Emirati gallerist, opened the Hunar Gallery in Rashidiya (Figure 3).

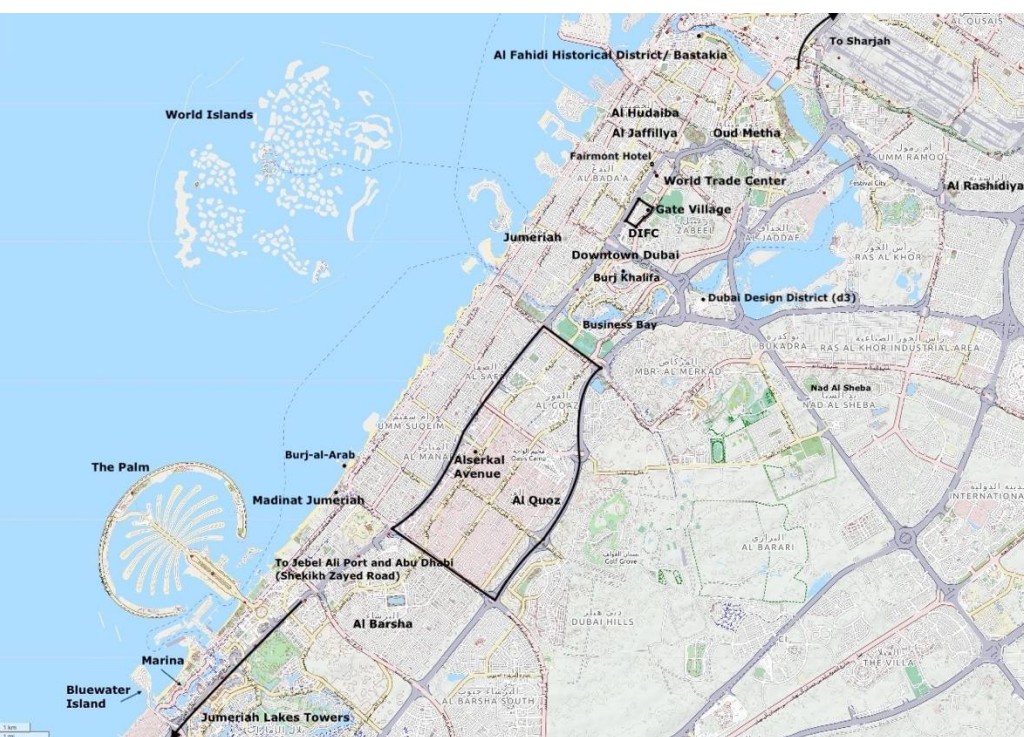

**Figure 3.** Map of Dubai indicating various neighbourhoods or buildings associated with the city's art market. Original map courtesy of OpenStreetMap.org, edited with permission by the author.

The 2000s witnessed an equally dynamic growth of galleries around Dubai, from the XVA Gallery in 2003 in Bastakiya to the Tabari Art Space Downtown in 2003, and in 2005 and 2006 the Meem Art Gallery, The Third Line, and B21 (now the Isabelle van den Eynde –IVDE—gallery), all in Al Quoz. The development of galleries in this area encouraged Hassan Sharif to open the Flying House in 2007; although considered more of an artist collective and exhibition space than a commercial gallery, it is nonetheless an example of a budding artistic climate (Allison 2017).

In the late 2000s, the establishment of larger exhibition spaces, though not commercial, contributed to the valorisation of local artists and galleries, helping the art scene flourish. These included the Dubai Community Theatre & Arts Centre (DUCTAC) in 2006, Traffic (an exhibition space), and Creek Art Fair (noncommercial, rebranded as Sikka Art Fair) in 2007, Alserkal Avenue (Alserkal) in 2008, and the Sharjah Art Foundation (SAF) in 2009. As for the commercial market, the first edition of Art Dubai (then known as the DIFC Gulf Art Fair) occurred around this time, as well as the opening of a branch of Bonhams in 2007, and the first local sale by Christie's in 2008. Magazines such as *Canvas*, *Bidoun*, and *Harper's Bazaar Arabia* were also founded to report and reflect on the burgeoning art scene.

While the presence of illustrious auction houses bolstered the UAE art scene—notably thanks to Christie's first auction whose sale of Farhad Moshiri's work for over a million dollars stunned the global art scene (Moghadam 2012)—other international ventures such as the Louvre and the Guggenheim announced their intention to open museums on Saadiyat Island, Abu Dhabi. The creation of such a cosmopolitan presence in the two largest emirates demonstrates a more international recognition of the local scene, which

continued to grow despite the 2008–2009 financial crisis. Despite this economic recession, the art market persisted, and galleries continued to open at hubs such as Alserkal Avenue (Figure 4) or the Dubai International Financial Centre (DIFC), and elsewhere in the city. Art Dubai (now with its current name, and new location at the Madinah Jumeriah hotel) continued to present a growing roster of galleries and programming every year; Alserkal Avenue announced a few years later a mirroring expansion, doubling the number of creative spaces, including international galleries such as Leila Heller and Stéphane Custot.

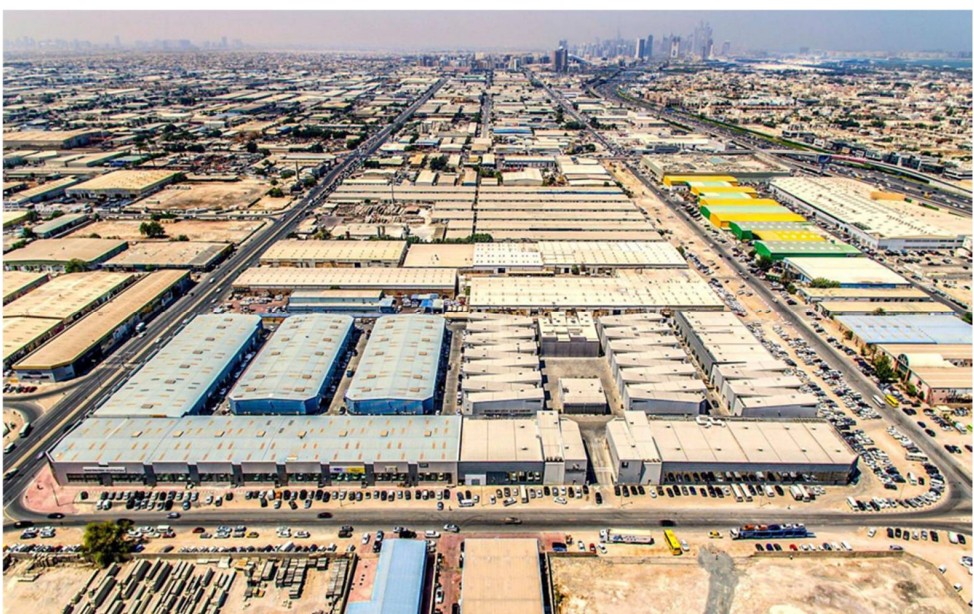

**Figure 4.** Aerial shot of Alserkal Avenue, 2017. Birds-eye-view of the art hub, with the original warehouses on the left and the newer expansion on the right. The view continues west towards Dubai Marina, over Al Quoz, with a view of Sheikh Zayed Road on the far right. Image courtesy of Alserkal Avenue.

This comprehensive pre-COVID timeline is necessary to demonstrate the astonishing rate at which the Emirati art scene, both commercial and nonprofit, grew from the early days of the country's history. Despite the lack of an established art history discourse, galleries, foundations, residencies, and artist collectives emerged, even though there was no existing infrastructure for aspects such as artistic or creative trade licenses, art handlers, art insurance companies, knowledgeable professionals to ensure correct light or climate control in exhibition rooms, etc. (Moghadam 2012). Indeed, in 2018, Antonia Carver, Director of the Dubai-based Jameel Art Centre and former director of Art Dubai, wrote an opinion piece on 'What Most of the World Still Does not Understand About the UAE Art Scene', in which she discussed the importance of adaptation and reinvention:

> The challenges we faced were often matters of infrastructure—how to build an audience in an emphatically diverse country of many languages, where many residents were unfamiliar with the district's galleries popped up in; how to establish nonprofits when the relevant government codes didn't yet exist; how to support young artists who were between residency visas. These challenges may have prompted invaluable creative collaborations, but frustratingly, many still exist. These include challenges with residency visas for artists who have a studio practice rather than a traditional 9-to-5; the lack of dedicated art supply specialty stores for working artists; the need for more higher education arts programs, plus under-developed archives for historic art practices. But each of these have been improving over the past decade and a new wave of arts governance is ambitious and intent on change. The growth of the scene in the UAE was, and is, messy

and non-linear, but it has a momentum that feels fresh and urgent, and a deep sense that the arts really matter. (Carver 2018)

The fast-paced nature of the UAE allowed those now considered art 'pioneers' to create a scene and market where none existed before. Doing so demanded a great deal of flexibility, as described by Carver. Each player involved in creating the market from the ground up had experienced what the 'art scene' meant, how it appeared, and functioned, elsewhere in the world (though not necessarily in a professional context). Being in the Gulf and not Europe, North America, or elsewhere in the Middle East, where there was already an art scene, those who endeavoured to cultivate the Emirati scene are exemplary of both the dynamic nature of the UAE and flexibility or open-mindedness, knowing they could never implant a preexisting art scene and history in the Gulf. As one simple example, earlier galleries were held in private homes rather than purpose-specific places. This flexibility does not exclusively mean financial flexibility—as most of the earlier galleries and artists funded their own projects and worked on the side—but rather flexibility of thought process, work habit, and the ability to break away from traditional art practices. Today, parts of the art scene do benefit from greater economic freedom but not, for example, the galleries that are the backbone of the UAE market. This flexibility served in their favor for such obstacles as the 2008–2009 financial crisis and has proven to do so again in with the COVID-19 pandemic. Rooted in the historical emergence of the Emirates' art world, this article outlines how the art scene's ability to adapt quickly allowed it to persist and retain a degree of normalcy during the pandemic. Many of the strategies employed by the different players in the art scene, in both the private and public spheres, were not unique in the world; what was unique, however, was how early on these approaches were implemented, and that, being a smaller market, in a country with a smaller population, nearly all actors adapted promptly and consistently throughout the different phases of the pandemic.

## 2. Chronology of the Pandemic in the UAE: Through the Lens of the Art Scene

Usually occurring in March, Art Week is the busiest time in the contemporary art calendar. Historically happening in Dubai, and revolving around its fair, this period has evolved to include events in Sharjah (Sharjah Biennale, spanning over several months, and March Meeting, over several weeks, as well as talks and exhibitions in that emirate's expansive variety of museums and foundations), and in Abu Dhabi, with shows at Manarat Al Saadiyat, Warehouse 421, NYU-Abu Dhabi (NYUAD) Art Gallery, the Louvre, etc. In Dubai, the schedule is even more robust with, for example, in 2019, three other art fairs occurring at that time (Fully Booked, Sikka Art Fair, World Art Dubai), and at least thirty-one exhibitions and events at art galleries, residencies, and foundations (Sartore 2019).

Art Week 2020 should have transpired similarly. The first cases of COVID-19 in the UAE were reported on 29 January and were the first in the Middle East (Gambrell 2020). A month before its opening date, on 24 February, Art Dubai was slated to go ahead as planned (Chaves 2020e). Indeed, most events that necessitated people gathering continued as planned: despite cases of COVID-19 rising in the UAE and the world, the *National* published an article entitled '66 things to do in Dubai, Abu Dhabi and the northern emirates this March', (National 2020a). New gallery spaces opened, such as the Oblong Gallery—originally founded in Forte Dei Marmi, Italy—on the new Bluewaters Island (Shehmir 2020b), and acquisitions at art centres such as Jameel (in the case of the Abraaj Art Prize collection) continued as usual (Gronlund 2020b).

Just a few days later, however, Art Dubai announced that the fair would not occur with the original agenda but would rather present a downsized version of events and talks, during the same dates, catering to a local audience. Chief executive Benedict Floyd, artistic director Pablo del Val, and international director Chloe Vaitsou said in a statement on 3 March:

> Given the essential role the fair plays in promoting local and regional artists, we have made the decision to stage a program tailored to the local cultural

community instead, including existing fair program contributors and thought-leaders. (Proctor 2020)

At this time, there were only 21 COVID-19 cases in the country (Gambrell 2020), but the directors reacted immediately to the growing numbers by simplifying the fair. Over the course of the next week, nearly every day presented the announcement of art events closing or being postponed: Sikka was deferred from late March to October (to coincide with Expo Dubai 2020), DIFC Galleries Night, scheduled for 23–24 March, was cancelled, through at the time the galleries remained open during normal business hours. In surrounding emirates, Warehouse 421 and NYUAD closed public programming and art spaces (and the latter preemptively announced reopening in early April), whereas SAF's March Meeting was delayed 'until further notice,' though the exhibition spaces remained open. The Louvre also closed its doors, initially for two weeks on the 14 March, only to be extended later; around this time, the museum also announced, 'the museum galleries may be temporarily closed, but our digital platforms continue to narrate our stories of cultural connections,' (Chaves 2020l). It should be noted that these closures were not government-implemented rules but the decision of the directors or the board of each institution or business, many of which were private, not governmental, entities. That being said, in the UAE, the lines between governmental, royal, and private are not always clearly defined. Governmental intervention to help the art scene survive will be discussed later, but the initiatives first taken by smaller, private groups should not be discounted.

At this stage in March, not everything was cancelled: Gallery Night in Alserkal Avenue (private) was still planned for the end of the month, and the galleries were not yet required to close; the non-profit spaces Jameel Art Centre (private) and Tashkeel (royally established but not run by the government) remained open (Chaves 2020f). Yet, on 14 March, Alserkal—despite intending to keep their programming as planned just a week prior—postponed or cancelled several exhibitions and their spring residency cycle. In the same statement, they revealed that the over 15 shows scheduled for Art Week would soon be viewable via online viewing rooms (OVR), providing prerecorded 3D tours (Gillet 2020). Art Dubai (majority private) followed suit, further altering plans for its 14th edition by moving all programming online: virtual performances, OVR, and a catalogue to replace booths, and the Global Art Forum talks broadcast live. For context, Art Basel Hong Kong—not surprising due to its proximity to the epicentre of the outbreak—and Frieze New York had also already completely cancelled (i.e., did not occur virtually) at this time (Chaves 2020d). Given the scope of this article, we cannot delve into detail on how Emirati art world entities reacted more quickly than their counterparts in different countries, though we will attempt throughout the article to provide examples of how the UAE did so earlier on and more consistently.

For the online fair, despite being virtual, the protocol to inquire about an artwork remained the same: interested collectors were still required to express their interest to the gallerist (through a form on the website), allowing for something familiar in these uncharted waters. We will later attempt to demonstrate that the evolution of this fair throughout the pandemic can be seen as representative of the entire UAE art scene's response to the pandemic: always adapting to new circumstances and always attempting to move forward. The online catalogue was rather complex and was searchable by artist, medium, and gallery, and specific works could be flagged as favourites. Thus, within a matter of days, the Art Dubai team succeeded in developing this platform, quickly adapting to the rapidly evolving pandemic. Another novelty for Art Dubai 2020 was that it refunded galleries half of the booth cost for this year, and granted a 50% credit for the 2021 fair, which they were certain would take place. Even the curators and subject of the Global Art Forum changed last minute to 'address the new stories emerging from this moment of global narrative collapse . . . collectively, with a group of brilliant thinkers, all of whom have something nourishing and provoking to say' (Shehmir 2020a).

Likewise, on 23 March, Alserkal Avenue launched its aforementioned virtual platform, allowing visitors to embark on a 360-degree tour of the exhibitions that were meant to

open for Galleries Night during Art Week. Beyond just a digital inventory of artworks, their tool provided details of the featured works, tags linking viewers to the website Artsy—where it was possible to purchase works online—and informational videos of the gallerists explaining the works on view, as if on a real gallery tour (Chaves 2020n). For the 'opening,' there were over 300 works from 15 different galleries and project spaces. Jameel implemented the same strategy by the end of the month (Chaves 2020g), as did all cultural spaces in Sharjah (Haza 2020). In the case of the latter, this was a government decision and not one of the private entities, as had been the case at art venues in Dubai and Abu Dhabi.

Thus, over the course of just one month, the UAE art scene changed how it presented itself many times—from more simplified programming to postponements or 'temporary' closures, to going fully digital. Both for- and nonprofit institutions remained resilient through the first month of the pandemic, setting the pace for months to come. Allowing for this resilience was their flexibility, which helped them react promptly to the unprecedented effects of COVID-19.

## 3. Adaptations: Private and Public Financial Support

Adjusting to the pandemic did not stop there. Beyond individual businesses (Art Dubai, and the various galleries) and nonprofit spaces (SAF, Jameel, Warehouse 421, the Louvre, etc.), the government announced at the end of March an initiative to purchase AED 1.5 million (over USD 400,000) of art from local artists (Binlot 2020). This level of financial support is unprecedented in the UAE for its art scene and also suggests an openness from the government to operate differently to survive the pandemic. First destined for Emiratis but later included long-term residents, the goal was to send a message of 'solidarity' to its artists. The art purchased would first be displayed in an online exhibition with Alserkal and then distributed to UAE embassies around the world. This initiative was conceived before the pandemic, but the urgency to support the arts presented the opportunity to launch it in late March (Dafoe 2020a). Hoping to support the entire art community, the UAE Minister of Culture and Knowledge Development (MCKD) Noura Al Kaabi revealed in an online Cultural Majlis on 1 April that freelancers and art businesses could benefit from government support as well. During this discussion with the founder of the Barjeel Foundation and co-founder of the Meem Gallery, Sultan Al Qassemi, she stressed the importance of 'adapting and modifying' their ways to see how the government 'can support the sustainability of the creative and cultural sector.' In this case, these are suggestions for the government to be more flexible—developing new ways of thinking about how to support the arts, beyond additional funding, but it should be reiterated that the arts sector, beyond this new government support, always had this kind of professional adaptability. Among certain measures discussed were rent and utilities relief and VAT exemption (Chaves 2020k), and they remained open to other suggestions.

The Ideathon was conceived in early April by the Dubai Culture & Arts Authority (DCAA)—a government entity—and Art Dubai—a private entity—to support the arts (Chaves 2020h). This online 'suggestion box' allowed anyone to submit ideas about how to support the local cultural scene, primarily with a focus on human capital, financial stability, the business community, and sustained creative production (Chaves 2020b). In addition, Al Kaabi's office also created, mid-April, a survey for the creative industry to ' . . . get a better understanding of the challenges facing the talent in the creative sector of the time . . . the next step will be to develop appropriate initiatives and incentives to ensure that the creative community is aptly supported to thrive' (Bedirian 2020).

Beyond these governmental initiatives, some real estate groups implemented rent relief policies (Chaves 2020k). Among them, Alserkal Avenue waived rent for its galleries for three months, as part of their 'Pay it Forward' initiative to help other local businesses and to try to keep operations, and salaries, afloat throughout lockdown (Chaves 2020o). Around this time, early April, the country's 'National Sterilisation Programme' was extended for another two weeks, with a strict 24 h lockdown: only one family member could leave at a time for necessities such as medical visits or groceries (Kell 2020). This was later

extended for another few weeks, only allowing businesses to open at the end of May (Harper's Bazaar Arabia 2020). While we cannot discuss in more detail how the Emirati government's sanitary and lockdown measures compare to that of all other countries, these were certainly stricter than that of the United States and most countries in Europe, for example.

Mid-May, Warehouse 421 (run by a royally established private foundation, though not a government entity) granted additional support through the Project Revival Fund. Open to regional 'mid-career visual artists, curators, writers, designers, and musicians,' recipients got up to AED 7300 (USD 2000), with funding available for at least 30 projects. The grants could be used for the costs of completing projects, renting equipment, conducting research, etc.—anything to keep the art and cultural world running. Jameel also announced a similar initiative, the Research and Practice Platform, providing funding for practitioners in the MENA region, putting aside a total of AED 550,000 (nearly USD 150,000) in grants (Chaves 2020m). In both cases, these are private, not governmental, initiatives run by Emirati art world pioneers (i.e., not necessarily native Emirati) that spearheaded these programs. In the case of Jameel, and perhaps also Warehouse 421, this funding came from the reallocation of the exhibition and event budget that was not being used at the time.

Despite this support for artists, galleries still suffered. The year 2019 had already been a difficult one due to the 2018 oil crisis. In Dubai, this meant less tourism, which makes up a large part of its wealth. While most tourists and—more recently, influencers—are not art collectors, the decrease in tourism had a negative effect on the art scene. As much of the arts programming happens in March, the timing of the coronavirus was particularly difficult.

This precarity was recognised by the government, specifically the MCKD, and by mid-May, the National Creative Relief Programme provided financial support to arts sector professionals such as freelancers, artists, and small businesses; the stipends ranged from AED 15,000 to 50,000 (approximately USD 4000–14,000). Spearheaded by the MCKD, the programme was a joint effort with the Corporate Social Responsibility (CSR) UAE Fund, allowing donations from individual and corporate sponsors, in addition to government support. This program was one of the results by the MCKD to the abovementioned survey launched in April, for which there were over 1400 responses. It is also perhaps because of the consequences of the 2008–2009 crash, in addition to 2019 being a particularly hard year for galleries in Dubai, that both public and private entities came together in so many ways to support this industry (Gronlund 2020a).

## 4. Unprecedented Collaborations: Response and Resilience

Collectively aware of their shared instability, several galleries joined forces to put together Not Cancelled Dubai: a week-long virtual event in May 2020. This digital fair featured OVR, talks, and tours (Chaves 2020j). The event was run by the Viennese Treat Agency, which inaugurated the initiative at home in April (Brown 2020) and then in other cities throughout Western Europe and the US (Artnet News 2020). The featured galleries were Carbon 12, Green Art, Grey Noise, IVDE, Lawrie Shabibi, and The Third Line, all of which are located at Alserkal (Figure 5).

In a comparable manner to how unconnected government entities came together to support the arts (i.e., the MCKD and the CSR Fund), this group of galleries—already closely tied by physical proximity and belonging to the same community—came together through this online fair with the aspiration to survive the pandemic together. Various gallerists expressed the importance of unity to confront:

> . . . the same set of challenges and disruption to our businesses. Being able to air those concerns has helped us all immensely—we are sharing resources, ideas and even staff . . . This is a turning point in our way of working . . . While nothing replaces the physical experience, the content that we will provide expands what you may probably get when you visit an exhibition at our galleries. (Chaves 2020j)

Later in 2020, another collaboration between the semi-private Art Dubai and the governmental DCAA, the Dubai Collection, emerged. The goal of the programme is to exhibit privately owned artwork—either from individual or corporate collections—throughout the city, instead of developing a national collection. Similar to the Art in Embassies programme, this scheme was in development previously, aiming to promote local collecting, thereby supporting local artists and galleries. This October announcement further demonstrated the UAE art scene's perseverance through the pandemic to continue past projects by adapting, rather than resisting, the effects of COVID-19. (Gronlund 2020f; Khaleej Times 2020). In a more in-depth study, it would be valuable to discuss the government response beyond its contribution to the art world during this time, though being limited in space, and because the government role is only a part of the art scene—especially the commercial one—such considerations cannot be taken at this time. Likewise, another pertinent addition to a longer work would include if and how there are examples in the past of a national culture authority partnering in such a way with an art fair elsewhere.

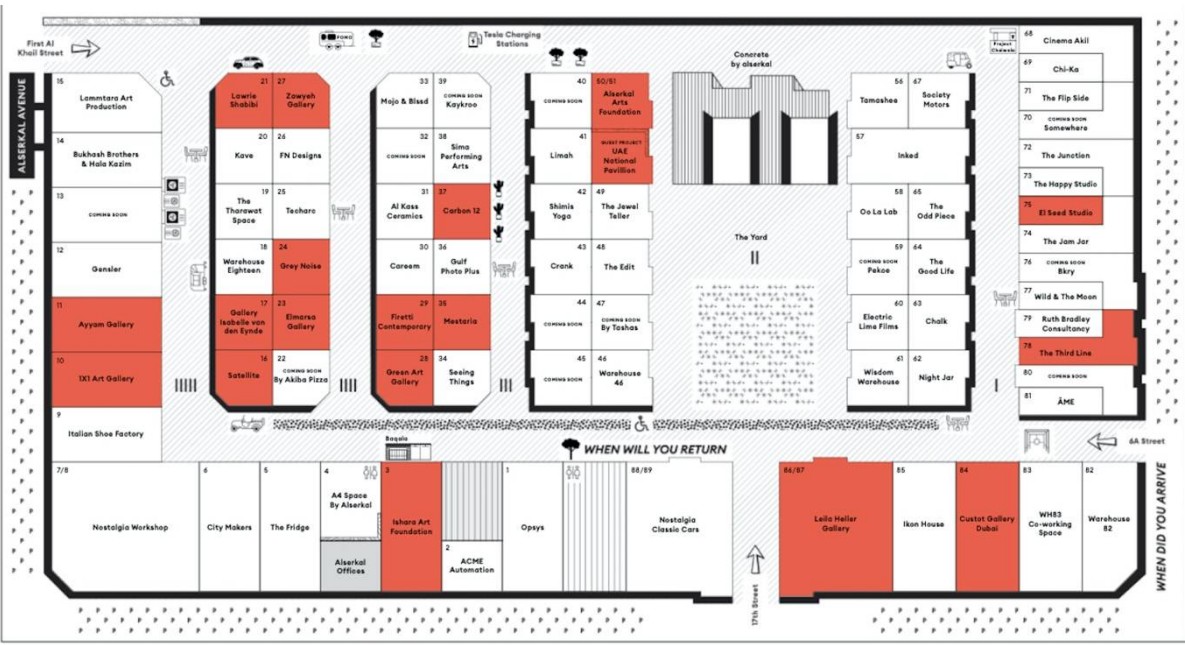

**Figure 5.** Map of Alserkal Avenue, 2021. Art spaces are highlighted in orange. Courtesy of Alserkal Avenue.

Along the same vein of earlier collaborations, and uncommon unity among otherwise competing ventures, one of the most apt examples of the UAE's art scene's resolve was a joint charity auction between Sotheby's and seven of the Alserkal Avenue galleries announced 11 June, scheduled for a week later. The funds were intended to support the Dubai galleries and contribute ten percent of the proceeds to the COVID-19 relief fund organised by the UN High Commissioner for Refugees. According to Will Lawrie, of Lawrie Shabibi, the community service encouraged by Alserkal's 'Pay It Forward' program further motivated people to give back. Entitled 'This Too Shall Pass: Bridging the Art World', The Third Line, IVDE, Carbon 12, Green Art, Leila Heller, 1 × 1, and Lawrie Shabibi contributed 63 lots from 45 artists for a total expected value of AED 2.9–4.4 million (USD 800,000–1.2 million).

Beyond this unlikely partnership, this sale also represents the first time many of the featured artists would participate in a Sotheby's sale. Moreover, the financial arrangements were also unconventional; the buyers' fee was maintained, and a portion of the selling price would contribute to the UN Relief Fund, but the commission from the seller (the gallery/artist) was waived, and the remaining revenue to be divided between the two of them (Gronlund 2020e). Of the 63 lots auctioned, 41 were sold, 6 were withdrawn, and

13 were not sold. Among those sold, 13 had a hammer price above the mid-estimate, and among those 5 sold at 100 percent more than the estimate (Mutual Art Auctions 2021).

Gallerists and auction house professionals on each side agreed that this kind of radical change was necessary; as Ashkan Baghestani, Sotheby's head of contemporary curated sales, stated:

> This is a time when people need to be open to new ideas . . . I'm happy that after years of trying to find different ways of engaging, finally the primary, secondary, and tertiary markets are interacting. The lines are a bit more blurred. It opens an infinity of possibilities in terms of partnerships. (Gronlund 2020e)

Lawrie agreed with Baghestani's remarks and added that the pandemic forced the art world to change established practices, such as pricing and budgeting, in order to survive (Gronlund 2020e). This is demonstrated the following year at Art Dubai 2021 and will be discussed later. By the end of June, despite being hindered by the cancellation of Art Week in March, some gallerists felt that life was 'business as usual', because, with the high temperatures, walk-in clients were always rare in the summer, and traditionally most sales were made with longstanding clients through outreach and email exchanges (Chaves 2020a).

> While there have been occasional instances of collaborations of gallery-gallery; gallery-auction house, they are infrequent. The fact that so many of these nearly exceptional partnerships took place in such a short period of time makes their existence more important, and further representative of the necessity of "thinking outside the box" to survive the pandemic. Due to the length constraints of this essay, we cannot go into detail regarding such alliances.

## 5. Inclusion of the Virtual Realm: A New Normal?

The digital strategies that were initially out of necessity and catalysed by the pandemic, have since been recognised as a valuable sales tactic, despite lockdown being over. Online galleries were on the rise before the pandemic but certainly have benefitted as a result; one gallerist described sales 'exploding' thanks to COVID-19 (Kewlani 2021). Furthermore, as the Emirates has an art scene that represents artists from regions less present in the Western market, collectors who seek art from the Global South or MENA region have a long tradition of forging strong relationships with local galleries. Thus, for example, when the Tabari Artspace was forced to launch a virtual version of Palestinian artist Hazem Harb's solo show—destined to occur in March—the show nearly sold out (Chaves 2020a). While there is no way of knowing what the results would be if the show were to occur in person, but given the sought-after nature of this artist's work, its success is of no surprise, and the gallery's quick reaction early in the pandemic to hold the show online foretells the local art scene's overall resiliency and adaptability.

Furthermore, such options were important because of how complicated international travel became. While the UAE opened relatively early, compared to other countries, artists, and artworks from elsewhere could still be delayed by the pandemic, as flights were subject to change and lockdowns could suddenly be announced (Chaves 2020a), and thus, the possibility of maintaining shows and sales online made sense financially. The UAE art scene has long been accustomed to fluctuations and forced to adapt in the past, helping it survive the pandemic.

While summer in the UAE is always slow, summer shows, and even new gallery openings, continued in July. The Ramallah-based Zawyeh Gallery, established in 2013, had its inaugural show at Alserkal in July 2020: *Palestinian Art: Resilience and Inspiration.* The title aptly demonstrates both the Palestinian resistance and the UAE's resistance against COVID-19 (Vakil 2020b).

In August, despite being far from its usual November date, Abu Dhabi Art published an online catalogue to further support galleries. The catalogue included over 200 artworks from 48 galleries which had previously participated in the fair. Dyala Nusseibeh, the fair

director, expressed the purpose of this programme being simply to help struggling galleries by offering connections with the fair's wide range of collectors during this difficult time:

> It is a simple effort to reach out to our collectors and visitors to encourage sales for these galleries. We hope to generate sales for galleries at a time when any extra support collectors can provide is much needed and appreciated. (Chaves 2020b)

The participating galleries were not charged to be featured in the catalogue, and the fair also announced an additional online campaign, Artist of the Week, to promote individual artists through pictures of their work and video interviews on the fair's website via social media (Chaves 2020b).

Other novelties that remarkably happened in August include Abu Dhabi's 101: more of a community than a gallery, 101's goals are to better represent younger UAE artists to create a more 'sustainable and ethical' art ecosystem. Aware of the fact that young artists often lack experience and know-how for pricing, this privately run platform holds quarterly sales of emerging and mid-career artists, splitting profit an uncommon 30/70, the majority going to the artist (Gronlund 2020d).

September harboured further announcements reflective of a thriving, though different, art scene. Warehouse 421 reopened after five months of closure. Despite this hiatus, the foundation remained in support of the arts through their relief fund, a residency program, a new podcast, and a focus-group initiative, in addition to continuing to organise the yearly SEAF exhibition. Appropriately telling of the UAE art scene's response to the pandemic, Faisal Al Hassan, general manager of Warehouse 421, summarises this moment clearly:

> The year of 2020 is the year of coping, and we're starting to look at 2021 as the year of experimentation. This is the new reality that we are living in. Even post-COVID, things will not be the same. We just need to be patient until we find the right formula where we can continue to have a physical space, go back to being social beings, but also making sure that the experience is right, with the digital and physical working together. (Chaves 2020c)

## 6. The Return to the Physical: Adapting to COVID-19 Safety Precautions

In the commercial realm, both Art Dubai and the less prestigious World Art Dubai announced plans to open in person for the next iterations during their regular springtime dates in 2021. World Art Dubai (private), which had postponed its event in March, revealed the next edition would occur in October 2020, at its usual Dubai World Trade Centre, with strict temperature screenings, ubiquitous gel-sanitising dispensers, and real-time tracking of crowd numbers and density. In addition, the fair set up contactless registration and ticket purchasing, as well as a partnership with fynd.art, an application designed to give visitors more information about an artwork, an artist, or galleries through a QR code (Rodrigues 2020), rendering physical, thus tactile, documents unnecessary. Returning to normalcy in 2021, World Art Dubai also held that year's version at the traditional time, following Art Dubai in early April as the region's more affordable fair. Boasting 250 galleries (Bedirian 2021) with over 200 artists from 27 countries, including, for the first time, Israel (National 2021), which in and of itself demonstrates progress despite the pandemic. The 2021 version, only a few months after the delayed 2020 event, was its biggest fair yet (Grover 2021).

Art Dubai also announcing both a new regional director, Hala Khayat, and a different format spread out throughout the UAE to celebrate the country's 50th anniversary during the 2021 fair. As another international comparison, while many fairs at the time, such as Frieze and Art Basel still resorted to the OVR model, Art Dubai planned to proceed with its programming, in an adapted format (Gronlund 2020c). This trajectory and determination are perhaps even more noteworthy considering that other large-scale fairs such as Art Basel Miami Beach and FIAC completely cancelled their events for later in the year (Dafoe 2020b).

Other artistic advancements continued in the fall. The private, members-only Arts Club, originally founded in 1863 in London, opened its doors in the DIFC area in November.

While not a traditional arts institution, this club fosters relationships between collectors, artists, and aficionados, and has a permanent collection of both established and emerging artists; it also features various cultural programming. Its existence in the UAE contributes to the growing arts community, and the fact that it opened a branch in 2020 is another example of how UAE art fared well during the pandemic (Bains 2020).

### 7. Returning to Normal, Continuing to Adapt

The UAE art scene's success through the pandemic relied on its adaptability—long part of its history—allowing the milieu to resume a somewhat normal functioning earlier than other countries. *Future Trends: Culture and the Creative Sector* (Dubai Future Foundation and Dubai Culture & Arts Authority 2020), published in October by the Dubai Future Foundation and DCAA, declared unsurprisingly, as this had already been the case, that the artistic and creative sectors require innovation in order to survive the pandemic. The collaboration between the two organisations was already novel, but beyond that, this partnership highlighted shared efforts already made (such as those between Sotheby's and Alserkal, or DCAA and Art Dubai) and presaged future efforts to come to support the art industry. Hala Badri, director-general of DCAA, aptly summarises this effort:

> In light of these exceptional circumstances, it is imperative that all those in charge of the sector in Dubai and the UAE intensify efforts and take measures to develop mechanisms and solutions to support the creative community and enable it to ensure its continuity and prosperity in the future, especially for small companies and independent entrepreneurs working in this sector. (National 2020b)

Though on the periphery of the contemporary art scene, the inaugural Dubai Design District (d3) Architectural Festival took place in November (Carpio 2020), as well as Dubai Design Week—both in person—further demonstrating the country's determination to persevere (Olele 2020). Abu Dhabi Art opted to hold its 12th edition mostly online, keeping in mind the difficulties that numerous other art fairs faced that year because of the pandemic, though some of the works were nonetheless viewable at their traditional location at Manarat Al Saadiyat.

By December, Art Dubai reaffirmed that it would launch as planned, in situ, in March 2021. It was the first major international fair to announce a physical event for the first quarter of 2021. The fair planned to present 86 galleries from 36 countries—over 90% of the galleries who sought to participate in the 2020 version. Artistic director Pablo del Val emphasised the importance of adaptability during the past year, yet also the desire to come together again for physical events (Thackara 2020). The organisers announced a new layout to allow for better crowd control and social distancing, in addition to hygiene measures that had become standard by then. At this time, the UAE had already vaccinated tens of thousands of citizens (Thackara 2020) allowing, in part, for Art Dubai to move forward as planned, while vaccination numbers stagnated in other countries that should have held fairs at the time. This prompted certain art critics elsewhere to doubt that having a fair so early on would be a good idea, or even feasible (Gerlis 2020a). In addition to its plans to open in person, Art Dubai maintained a visual presence online through its Portrait Exhibition, focusing on individual artists from the Global South, with works available for sale. The exhibition was scheduled to run from late December 2020 through 20 January but was later extended through February 2021 (Art Dubai Online Exhibitions 2020; Vakil 2020).

For galleries willing to participate in the fair but unable to travel, the organisers created the Remote Participation Programme: this entailed the Art Dubai team hanging works according to the gallerist's plans and have staff occupy the booths to coordinate between the collectors and the galleries. The fair also revealed a revolutionary sales model: rather than paying for the booth upfront, galleries would reimburse Art Dubai 50% of their profit after sales, up to the cost of the booth. Further anticipating hygiene measures, the fair also designed an application to allow visitors to book tickets directly without having to interact with staff in person (Dawson 2021). Sanitary measures were taken very seriously by Art Dubai, and it also helped that the UAE maintained one of the best vaccination rates

in the world, second after Israel (Said and Faucon 2021). These numbers are of course thanks to the governmental response and not that of the art world, but they nonetheless contributed to a relative return to normal in the cultural sphere. Art Dubai also offered free storage for three months after the fair, taking into consideration augmented shipping delays due to the pandemic, and allowed galleries to postpone their participation last minute to the 2022 edition (Brady 2021).

As many Western countries had stricter travel regulations at the time—in addition to fewer vaccinations with higher infection rates—most galleries represented were from the Global South. While Art Dubai had for a few years endeavoured to benefit from its central geographic location and establish itself as a gateway between cultures, the consequences brought on by the pandemic expedited this process (Gronlund 2021g).

Elsewhere in the UAE, galleries and museums continued to host physical shows with social distancing and sanitary regulations, while new ventures continued to be announced: NYUAD revealed the country's first MFA program (Gronlund 2021h) to open in the fall of 2021. This was an important advancement as collectors had previously expressed a lack of approval of the UAE art scene since there were no art schools (Proctor 2021b).

In March 2021, Art Dubai officially became the first major fair to launch since the start of the pandemic, after being one of the first to close a year prior. In terms of programming, the traditional curated sections (Modern and Bawwaba, focusing on individual artists from the Global South) were not realised but were supplemented by an outdoor sculpture park and a greater emphasis on video art, with screening locations spread out throughout the fair (Chaves 2021a). Food venues, VIP lounges, and other social activities were eliminated, reducing the event to its primary goal—to sell art. Executive Director Benedetta Ghione, said in an interview:

> We thought, if we can't have it all, then let's strip it down to the core and keep the essentials. How are we trying to make a difference here? Ensuring the sustainability of galleries and artists in these difficult times was and is our most essential goal. (Selections Arts Magazine 2021)

The final numbers show that 50 galleries from 31 countries were present, a smaller group than the 2019 iteration, which hosted 92 galleries from 80 countries; however, many visitors preferred the more simplified layout and the new location in the more central DIFC. Galleries were also pleased, since many, such as Leila Heller, Custot, Meem, Perrotin, Templon, Comptoir des Mines, Galleria Continua, and Gallery 1957, all reported successful sales (Proctor 2021a, 2021b). The first public day alone had sales of up to three million dollars (Chaves 2021b; Batycka 2021), demonstrating collectors' desire to return to in-person purchases, as well as the fair's unexpected success. The final sales amounted to 'over three million dollars . . . in line with pre-pandemic figures' (Said 2021). Despite the stagnation caused by the pandemic, the fair occurred at a time of change for the UAE: in September 2020, the Abraham Accords were signed with Israel, and the embargo placed on Qatar (whose ruling family are major collectors) was lifted (Proctor 2021b). Just after the fair, the ruler of Dubai, Sheikh Mohammed bin Rashid al Maktoum, also loosened restrictions during the holy month of Ramadan, allowing restaurants to serve customers during fasting hours, whereas before establishments, they had to either close, or put up curtains, dividers, or facades (Government of Dubai 2021).

While Art Dubai succeeded in taking place in person, many other international fairs—The Winter Show in New York; BRAFA in Brafa in Brussels; Tefaf Maastrict; Art Basel Hong Kong; Freize Los Angeles; ARCO in Madrid; and Art Rotterdam—remained online or postponed their dates to later in the year (Brady and Jhala 2021; Reyburn 2021). The fact that it was one of the first to cancel in 2020 and yet the first to take place in person in 2021, and boast lucrative results, exemplifies the Emirati art scene's overall successful handling of the pandemic. Beyond the fair, other events such as SAF's March Meeting were maintained and institutions such as the Louvre, Jameel, Sharjah Museums Authority, Alserkal, SAF Warehouse 421, NYUAD Art Gallery, and Abu Dhabi's Cultural Foundation

(i.e., much of the UAE art scene) also held physical shows and programming while still adapting to the new circumstances (Chaves 2020i; Chaves 2021c).

Progress continued elsewhere as well: in April, Alserkal Avenue revealed new goals and a different artistic direction. These measures included an art advisory programme to encourage 'public and private sector entities in developing sustainable and responsive business models', a sustainability plan, and Alserkal.online, a 'digital platform led by the creative community, as a forum for cultural discourse, digital art commissions and multidisciplinary editorial content' (Yusuf 2021).

The public sector announced new cultural projects that same month. Sheikh Mohammed revealed plans to bolster the arts with an objective to increase the gross domestic product from around two to five percent and to double the number of creative businesses by 2025 (Reynolds 2021). A few days after this announcement, as the first manifestation of this goal, Sheikh Hamdan bin Mohammed bin Rashid Al Maktoum, the Crown Prince of Dubai and Chairman of the city's Executive Council, declared plans to transform Al Quoz into a creative zone for artists and designers (Arabian Business 2021). While this project is nascent, the campaign for realising these ambitions continues to unfold.

Within one year, the Emirati art scene succumbed, as did everywhere, to the devastation of COVID-19, but thanks to this scene's ability to innovate, renew, and adapt, it has emerged stronger than before. The fact that the timeline of events and notable moments since March 2020 is entirely filled is a testament alone to the scene's resiliency. This phenomenon is partly due to the flexibility, from its origins, that all actors in the art milieu had to possess to adapt to, rather than resist, the pandemic. The fast-paced creation and evolution of the UAE art scene, present since its inception and outlined in the introduction, also contributed to its survival. Beyond that, the arts scene, that of Dubai especially, has a tradition of persevering through sudden forced change and economic woes. In another comparison to other countries, the creative sector in the UAE strongly came together during the pandemic, while at this time in France, the Comité Professionnel des Galeries d'Art sued the government for not being allowed to remain open during the spring 2021 lockdown, even though auction houses could (Rea 2021). It was perhaps easier to create unity among the different players in the UAE art scene thanks to its lack of historical tradition and smaller size.

## 8. Conclusions: Possibilities in a Post-Pandemic World

Keeping in mind the timeline and success of Art Dubai as a microcosm of the country's art scene, this also reflects the entire art scene's adaptability. Before the pandemic, the fair had already undergone major changes in leadership, funding, and activity. After several years under Antonia Carver, Myrna Ayyad became the director in 2016. After only two years, she stepped down and, initiating further change, the fair altered its leadership structure entirely to have multiple directors. In 2019, Art Dubai was forced again to adjust: Its long-term and most important sponsor, the Abraaj Group, filed for liquidation and pulled all funding (Shaw 2018). In addition to these vicissitudes, for nearly every edition, new programmes were implemented, or other organisational changes were made. The impact of COVID-19 was thus just another, albeit more extreme, change for Art Dubai, and seeing how it, along with the rest of the UAE art scene, adapted to the pandemic reflects both its past and suggests how it will evolve in the future. Many gallerists agreed that Art Dubai met all the challenges and necessary changes for this year's fair and that with a 'clear vision and a lot of flexibility', it can 'meet the demands of the market and of the local and regional scenes' (Estiler 2021). One pertinent example is the post-fair payment system. While Ben Floyd said that this was going to be a one-off practice due to the pandemic, he did not exclude the idea of something similar in the future (Kerr 2021). Gallerists such as Sunny Rahbar of The Third Line understand that this model may not be possible for all fairs going forward, but she did express the desire for it to be considered:

> I wish more art fairs considered it, basically the fair is taking the risk that normally the galleries take upfront. It's quite a novel proposal and I realise it's likely not

> possible for other fairs to adopt this model but why shouldn't this be considered
> or at least something more along these lines versus standard models ... The
> world has changed and so will the models of engagement whether financial
> or social. I think we all need to adapt in order to continue to do what we do.
> (Brady 2021)

Looking onward, certain actions and adaptations that parts of the UAE art community made during the pandemic may presage how the international art world will continue to evolve in the future. For one, art fairs may be reduced to smaller, more human-scale events. Even in early April 2020, articles discussed the possibility of the pandemic having the benefit of lessening 'fairtigue'—in 2000, there were 55 art fairs globally, and 300 in 2019. While little was known about the outcome of the pandemic or the future of the art scene, early on, one author predicted a smaller number of fairs (Adam 2020). Nearly a year later, many fairs have been cancelled, and for those that have been realised, they are either far reduced in size, online, or a hybrid of both. Frieze New York (5–9 May 2021), moved from Randall's Island to a smaller venue in Hudson Yards with less than a third of exhibitors than the 2019 event, and it also opened its OVR to over one hundred galleries through 14 May. Other fairs continue online: Art Basel Hong Kong planned to launch at the end of May with art present, but no dealers, and have a remote booth system similar to Art Dubai, but it was later postponed; 1–54 Art Fair took place in May in New York, though most of the works were shown online (Gerlis 2021). While many gallerists agree that online viewing is not the best way to experience art, it has saved many galleries, and thus artists, during the pandemic; the global art scene in the future may present a combination of both smaller fairs, some with an online option, but also the possibility to visit in person (Gerlis 2020b). This article did not attempt to put forth the notion that none of the measures taken in the UAE were taken elsewhere, but rather that they were executed more promptly and in a more widespread manner. We fully acknowledge that part of this is due to the smaller size, population, and wealth of the UAE, but as this essay focuses on that country as a case study, we have endeavoured to give an in-depth view of how it reacted—and survived—the pandemic.

A greater online presence may also manifest itself for auction houses and galleries as well. In the first days of lockdowns in March 2020, Sotheby's launched its first online-only auction for Modern and Contemporary Art from Africa and the Middle East, making over GBP 2,193,625 (USD 2,679,294), 'with sixty percent of lots exceeding their pre-sale high estimates' (Sotheby's 2020). In March 2021, while the UAE art season kicked off, the online marketplace Artsy launched two initiatives focusing on the region: Middle Eastern Galleries Now, an online fair-like presentation of galleries from the region, and Legacy Trilogy: Past, Present, Future, a fundraising exhibition. The former ran for several weeks in March, and while some galleries before such as 1 × 1 and Carbon 12 first resisted the platform, many are now open to it as collectors adopt this technology. As mentioned earlier, this also reflects the rise in online galleries. Legacy Trilogy, organised in partnership with the nonprofit ArteEast, was designed to replace that organisation's annual benefit auction to raise COVID-19 relief money; this philanthropy may have been inspired by Sotheby's charity auction in summer 2020 (Chaves 2021d). As the world inches out of the grips of the pandemic, the art scene of yesteryear will not return, but some of the changes outlined in this essay may illustrate what is yet to come.

**Funding:** This research was funded by the Institut d'histoire moderne et contemporaine (IHMC), UMR 8066; Artl@s (Ecole normale supérieure—PSL); and the Bourses Mobilité Île-de-France, notably for two trips to the UAE in January 2020 and March/April 2021, the latter of which allowed the author to gain a first-hand experience with the post-COVID-19 art scene in the UAE.

**Institutional Review Board Statement:** Not applicable.

**Informed Consent Statement:** Not applicable.

**Data Availability Statement:** Not applicable.

**Acknowledgments:** The author wishes to thank her thesis director Joyeux Prunel for her ongoing support for this project, as well as Sophie-Rose Schor (Doctoral Candidate at the University of Massachusetts-Amherst) for her diligent editorial contributions. This project would not be possible without the time and generosity of those interviewed. She would also like to thank the team at Alserkal Avenue for providing her with archival information from its early days, and the attentive librarians at the Jameel Arts Centre.

**Conflicts of Interest:** The author declares no conflict of interest.

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
