# Peer review of "Global Art Market in the Aftermath of COVID-19: A Case Study on the United Arab Emirates"

_arts, 2021_

Round 1

Reviewer 1 Report

This is an interesting article that provides a sound introduction to ways in which the arts scene in the UAE responded to the COVID-19 pandemic. Emphasis is placed on the speed with which art fairs and gallerists based in the UAE adapted to the new circumstances, swiftly moving their operations online. Information is also provided on the types of state support offered to local artists and artworld professionals/institutions for the purposes of weathering the pandemic.

The article also offers a good history of the emergence of the art infrastructure of the UAE, and I am sure that this would be a useful guide to many readers who are unfamiliar with the cultural development of region. 

Generally, the article is well written and has a good structure. There are two important issues that need to be dealt with in order to bring the article to a publishable standard. The first concerns the theoretical framework of the discussion. The article is largely descriptive, and this is reflected in the bibliography. The latter is almost entirely comprised of newspaper/art journalism articles. These provide evidence (or journalists' opinions) of various developments in the art environment under discussion, but the discussion never broadens onto, or event considers, the wider theoretical debates or ideas that are shaping the world of contemporary art, art fairs, or art markets. The result is that the article is theoretically under-developed.

The second issue concerns the central claim of the article, i.e. that the UAE arts scene was able to react more swiftly and effectively than other nations to the pandemic because of the UAE's unique cultural history (in particular, its creation of an arts infrastructure in a matter of decades). The art market in the region is, therefore, posited as being uniquely adaptable.

These points are asserted in the article, but they are not proved. Part of the problem is that the article treats the UAE arts scene in a way that is entirely separate from its politics, social, and financial structures (including the ability of the local health system to respond to the pandemic, the wealth of the Emirates, and the small numbers of population involved by comparison to other countries). Substantiating the central claim would need to include more detailed consideration the role that art plays in the region as a major tourist attraction and business operation – and, hence, source of income. These are issues that give local rulers a vested interest in supporting the arts. The article concludes that the in-person staging of Art Dubai in 2021 'reflects the Emirates' ability to adapt', but this is not substantiated in the body of the text.  

If the argument is to be sustained that the UAE arts scene adapted to the COVID-19 situation better than other countries, more detailed comparison with responses to/levels of government support for the arts by other countries is needed (state support for the arts was offered by many countries). Other art fairs, auction houses, and museums swiftly moved to online offerings at the outbreak of the pandemic, so further detail is needed on what was distinctive about the activities undertaken in the Emirates.

While there is potential in this article, the central claim needs to be more fully evidenced, contextualized, and defended. 

Author Response

Dear reviewer,

Thank you for the time you took to read this article and for well-thought out comments. You are completely right in that the article lacks a theoretical basis. This was done concssiouly, as I hope to have adressed in the introduction. Given the constraints of the article length and the fact that, in my opinion, part of what made the UAE art scene's response to the pandemic unique was how the art world still continued with events and programming despite not being able to function like before. To demonstrate this, it was necessary to go through the timeline of events, taking up a great deal of space. These events and this timeline was favored over past theoretical issues that were produced before the pandemic.

I hope to have made the idea of the UAE art scene's adaptability more clear in this second draft. I have attempted to provide more explanations with the government response as well as comparisons to other art scene of other countries. However, I did not shift the focus to discuss the art scene's relationship to the country's politics, social and financial structures, as the majority of the art scene is privately run and not only reflective of government measures.

I am delighted to keep these comments and ideas in mind for future research and publication opportunities that will allow to to explore such issues at greater length.

Best,

Eve Grinstead

Reviewer 2 Report

This is a very thoroughly resourced article which clearly documents a chronological response by the UAE arts industry to the global pandemic.  It is well structured with a clearly defined methodology. The article draws on an extensive range of primary sources. Overall, the content of the article logically supports a compelling argument.

I don't see the need for any revision to this article. However, the article could be further enhanced by the identification of a couple of key outcomes in order to further support the author's argument, and offer extended critical analysis. I highlight these key points here, whilst acknowledging that this information may not be available to the author due to the recent nature of these events:

  1. Section 4 (article lines 291-298) - the author refers to 'Unusual Collaborations'. Had there been any history of collaborations of galleries prior to the pandemic? Why does the author propose that they are unusual?
  2. Sotheby's/Alserkal auction (article lines 342-349) - what was the result of this sale? Was it a financial success?
  3. Hazem Herb's online exhibition (article lines 360-361) was a sellout; however, would this have been the case if it had been an IRL event?
  4. Art Dubai's 'profit after sales' arrangement (article lines 518-539) - how successful was this in the end for Art Dubai? Did they make enough money? How did their income from this arrangement compare to how much they would usually earn from selling booths?

Finally, as a small copy-edit matter, I would suggest that the author check the verb tenses throughout. In a few cases, the author uses a present tense verb, whereas a past tense would clarify the historic nature of the matter being discussed, particularly considering this is a chronological methodology.

Overall, I commend the author on a comprehensive, interesting and well-constructed article.

Author Response

Dear Reader,

Thank you for the time you took to read this article and for your fruitful comments.

My comments:

  1. I hope to have clarified what I meant by "unusual" collaborations and to have provided insight on the historical importance of these alliances.
  2. I was going to include this in the first draft and didn't think it would be necessary--thank you for reminding me to include them.
  3. I hope to have replied to your question about Harb's virtual vs. IRL show.
  4. I hope to have replied to your question about Art Dubai's new post-sale payment system.
  5. Thank you for the grammar reminder, noted!

Best,

Eve

Reviewer 3 Report

The article dwells on a timely issue, discussing the case of the art scene in the UAE. It is extremely relevant to current discussions on the coping of particular art arenas with the Covid-19 crisis and provides food for thought on possible strategies that can contribute to the resilience of the art market at a time of crisis and in general.

The piece is written in a clear style and language, the structure serves the aims and the argument is articulated coherently. I do, however, find some flows, and I hope my following suggestions can help improve the paper:

The main problem, to my mind, is that the argument is not strong enough. The writer demonstrates the exceptional promptness of the UAE art arena in adapting to Covid-19 consequences, and attributes this promptness to a quality - flexibility. Methodologically, this course of argument is problematic for two reasons: first, the author does not define flexibility as a concept or as a practice; the paper elaborates on the quick growth of the field prior to the pandemic, but this alone does not imply flexibility (of what -thought? finance?). Second, the paper neglects a very important aspect of the described issues - politics and government policy.  Throughout the article, new initiatives, financial support, and operative decisions are presented as a strategy developed and adopted by the art arena itself, while the reality is far more complexed, and should be contextualized in other sectors and fields as well. As the author shows in lines 564-567, other governments did not allow offline events - and this weakens the claim that flexibility was what helped the UAE art field endure successfully.

Roughly speaking, my suggestion is to shift the focus of the article to the political aspect, elaborating on state interests and goals. By showing how pandemic-management strategies were designed in the UAE according to a political agenda (international affairs, economic considerations, and so on), the author will be able to make a stronger argument about the specificity of the UAE art arena. This arena was effected by: (a) government policy - that opened the borders and minimized restrictions (due to political interests); (b) the habituation of this field to working for local audiences and for foreigners who were used to buying online; (c) a relatively developed infrastructure that enabled quick online solutions and collaborations. In its current shape, the article addresses only the third aspect (and briefly passing by the second). I believe that an analysis of the first aspect will enable a much stronger claim, and will result in a study, which contribution to the field of discourse will be higher and significant.

Another point is that this work can benefit from an engagement with the academic literature on the art field in the gulf region. The past decade saw exponential growth in studies recognizing the centrality of the gulf area to the world art scene, trying to construe gulf-countries strategies with regard to art. To name just a few sources that I recommend engaging with: Robertson, Iain. 2011. A new art from emerging markets. Surrey, UK: Lund; Humphries. Erskine-Loftus, Pamela, Victoria Penziner Hightower, and Mariam Ibrahim Al-Mulla. 2018. Representing the nation: heritage, museums, national narratives, and identity in the Arab Gulf States. London and New York : Routledge Taylor & Francis Group.; Lord, Gail Dexter, and Ngaire Blankenberg. 2015. Cities, museums and soft power. Washington, D.C. : American Alliance of Museums Press. 

Finally, it will be good to incorporate quantitative data on sales and turnovers. the author should make clear why the art sector in the UAE is different than other sectors (if indeed). Also, since we now know that the Covid-19 period resulted in a growth of the upper class - the very class that nurtures the art market - why is it strange that the UAE art sector was not hurt? (perhaps if the author shows better results in comparison to the art sector in other countries, the claims will get more convincing).

To sum up, I find this piece interesting and timely, but I recommend further work in order the design convincing arguments and significant discursive impact.  

Author Response

Dear reader,

Thank you for the time you took to read this article and for your detailed response.

I hope to have made sufficient changes according to your suggestions and critiques. I have endeavored to define more clearly the idea of "flexibility" and why that is important. I have also tried to address the issue of politics and government policy--partly by pointing out that most of the measures taken and responses to the pandemic were not governmental, but private. If the article were addressing only non-profit organizations like the Louvre or the SAF, perhaps my point of view would be different, but being that this article discussed the art market, thus commercial and made up exclusively of private entities, I did not focus on policy.  

As for including more academic literature on the Gulf, there was not enough space to do so and give a history of the arts in the UAE, provide a detailed account of how the local art scene responded to the pandemic and occasionally compare these actions to that of other countries' art scene. Furthermore, as this article aimed to focus on the UAE art scene's response to the pandemic, which is still unfold, I found that current research did not aptly treat the subject, hence why I relied on first hand field work and press articles.

I will keep your just comments in mind for future work, and thank you again for your time.

Best,

Eve

Round 2

Reviewer 1 Report

Methods and approach have been helpfully clarified. The reasons why the UAE are understood as forming a uniquely adaptable arts environment have also been further elucidated. Together, these changes help to position the research and to highlight the contribution of the paper to scholarship in this area.

The article is an interesting and enjoyable read. It requires a round of copyediting to eliminate minor typos etc, but is otherwise ready for publication.

Author Response

Dear reviewer,

Thank you for taking the time to read this again. As my first peer-reviewed article, this (as you may have seen) was a challenge for me. I am content with the final result given the time constraints, but hope to improve in the future. Your comments were very helpful for making the necessary changes.

Best,

Eve

Reviewer 3 Report

It seems like the writer made an effort to address some of the problems I indicated in the first round. The two major points that were revised are: (a) an explanation of the term "flexibility", employed by the author as a methodological concept that ultimately helps build the argument. (b) a declared limitation of the article to a descriptive form of the online functioning of the galleries, avoiding, intentionally, a comprehensive analysis that relates to political concerns.

I still think this article carries more potential for merit, that its current shape does not reflect, but I suppose within the limited time frame given for review, this is as good as it gets.  

Author Response

(The authors gave the same response as above.)
